# Revisiting Alkali Metals As a Tool to Characterize Patterns of Mosquito Dispersal and Oviposition

**DOI:** 10.3390/insects10080220

**Published:** 2019-07-24

**Authors:** Matteo Marcantonio, Olivia C. Winokur, Christopher M. Barker

**Affiliations:** Department of Pathology, Microbiology, and Immunology, School of Veterinary Medicine, University of California, Davis, CA 95616, USA

**Keywords:** invasive species, *Aedes aegypti*, mark-recapture, species dispersal, alkali metals, vector of pathogens

## Abstract

Mark-recapture methods constitute a set of classical ecological tools that are used to collect information on species dispersal and population size. These methods have advanced knowledge in disparate scientific fields, from conservation biology to pest control. Gathering information on the dispersal of mosquito species, such as *Aedes aegypti*, has become critical since the recognition of their role as vectors of pathogens. Here, we evaluate a method to mark mosquitoes that exploits the rare alkali metals rubidium (Rb) and caesium (Cs), which have been used previously to mark adult insects through feeding. We revised this method by adding Rb and Cs directly to water in which the immature stages of *Ae. aegypti* were allowed to develop. We then assessed the effect of Rb- and Cs-enriched water on fitness, survival and bioaccumulation in both adult females and their eggs. Results indicated that Cs had adverse effects on *Ae. aegypti*, even at low concentrations, whereas Rb at low concentrations had no measured effects on exposed individuals and accumulated at detectable levels in adult females. The method described here relies on passive uptake of Rb during immature stages, which has the benefit of avoiding handling or manipulation of the dispersive adults, which enables purer measurement of movement. Moreover, we demonstrated that Rb was transferred efficiently from the marked females to their eggs. To our knowledge, Rb is the only marker used for mosquitoes that has been shown to transfer vertically from females to eggs. The application of Rb rather than more traditional markers may therefore increase the quality (no impact on released individuals) and quantity (both adults and eggs are marked) of data collected during MR studies. The method we propose here can be used in combination with other markers, such as stable isotopes, in order to maximize the information collected during MR experiments.

## 1. Introduction

As early as the late 19th century, mark-recapture (MR) methods have been applied to estimate dispersal, survival and population sizes of numerous animal species [1]. Insects were among the first taxa to be subjected to MR studies [2] and among these, mosquitoes are well-represented [3] due to their importance as vectors of pathogens. During the long history of mosquito MR studies, several marking techniques have been developed, nearly all of which have targeted adult mosquitoes [4].

A major drawback of most marking methods for adult insects is that the marking process may modify fitness and behavior compared to unmarked individuals [5]. To overcome this issue, recent studies have artificially altered the stable isotopic signatures of carbon (δ^13^C) and nitrogen (δ^15^N) in the water in which mosquito larvae develop [6]. Container-breeding mosquito larvae reared in water with altered δ^13^C or δ^15^N emerge as isotope-marked adult mosquitoes due to bioaccumulation of the carbon and nitrogen isotopes in tissues as larvae develop. This procedure avoids the manipulation of adults during the marking process, thereby minimizing changes in behavior and fitness of the released individuals.

Despite these innovative and beneficial features, stable isotopes as markers are highly sensitive to contamination and do not persist at detectable levels into the next generation to allow tracking of oviposition by marked females [6]. For this reason, we propose a complementary marking method to increase the information gained from MR experiments.

Alkali metals represent a distinct group of elements in the periodic table, characterized by low ionization energies [7]. Potassium (K) and Sodium (Na) are the most abundant elements in this group and both have a critical role in biological systems, particularly in ion exchange and osmotic balance within cells. Rubidium (Rb) and caesium (Cs) are two alkali metals that are rarer than potassium (K) and sodium (Na) in the Earth’s crust but with physical and chemical characteristics very similar to K. When Rb and Cs are made available to organisms, they indeed follow the same physiological path of K, except that they displace K and accumulate in the intracellular fluid, due to their higher affinity with fixed negative charges [8]. The rarity in nature, low toxicity and bioaccumulative properties of Rb and Cs make them valuable as markers for MR studies.

Rb and Cs have been used repeatedly to mark insects. In 1972, Rb was first proposed as a marker for insects [9] and thereafter was successfully used in diets containing 100–10,000 mg L^−1^ of Rb to mark tobacco budworms [10]. Rubidium and Cs have been injected into vertebrate hosts to track the blood-feeding behavior of adult *Culex tarsalis* and *Culiseta melanura* [11,12] and to investigate the dispersal of the biting midge *Culicoides varipennis*, a vector of animal pathogens [13,14]. Rubidium delivered via bloodmeals has been used as a marker to study the oviposition patterns of *Ae. aegypti* in an urban area [15]. More recently, other authors employed Rb-marking to track the dispersal of *Ae. aegypti* in tropical urban areas [16,17,18]. All the listed mosquito MR studies involved the release of adult females (2–6 days old) fed with blood enriched with Rb before release, excluding male mosquitoes from the study. Moreover, these studies have been conducted with mosquitoes reared under controlled environmental conditions and handled by investigators, potentially impacting their fitness and thus their dispersal.

Evidence suggests that mosquitoes and other insects that are exposed to concentrations of Rb or Cs above natural levels during the larval stage will emerge as adults with a high concentration of these elements within their tissues and are thereby “marked” [19]. We hypothesize that exposure of immatures to a sufficiently high concentration of Rb and Cs would result in vertical transfer of these elements from marked adult females to their eggs. In addition, we postulate that elevated concentrations of Rb and Cs in the larval habitat should result in high concentrations in ovarian follicles and eventually eggs, as in mosquitoes, ovaries are formed by the 4th larval instar [20]. We expect that the concentration in eggs would be lower than that of the ovipositing females due to continued physiological turnover and ovarian development post-emergence.

Building upon these premises, the general objective of this study was to test whether Rb or Cs exposure during larval development yields reliable marking through accumulation in adult mosquitoes and whether the marks are transferred to the eggs. We also sought to estimate the optimal concentration of Rb and Cs in larval rearing water to produce reliable marking for MR experiments without affecting fitness and survival [19]. We used the yellow fever mosquito, *Ae. aegypti*, as model organism due to its importance as an invasive pest and vector of human pathogens [21]. The success of *Ae. aegypti* control programs depends on an understanding of the species’ potential for dispersal [22,23], which can be evaluated through MR experiments.

## 2. Materials and Methods

### 2.1. Mosquito Rearing

All experiments were carried out using colonized *Ae. aegypti* eggs (F4 and F14 in successive experiments) from individuals collected in Los Angeles County, California. Eggs were selected for hatching only if they appeared to be viable (undamaged, unhatched and without mold or discoloration) during inspection under a dissecting microscope. Hatching medium was prepared by adding 0.18 g of Oxoid Broth (CM0001, Thermo Scientific, Hampshire, England) powder to 500 mL of deionized (DI) water, together with 0.5 g of brewer’s yeast in 4.92 L plastic trays stored in an environmental chamber at 25.6 °C and 80.0% relative humidity. 99% RbCl and CsCl (Chem Center, La Jolla, CA, USA) were added to this solution to obtain Rb and Cs concentrations ranging from 0 (hereafter referred to as “control”) to 1000 mg L^−1^. After 24 h, the eggs were added to the trays and submerged. Finely ground fish food flakes (TetraMin, Tetra, Blacksburg, VA, USA) were added daily to each tray at a rate of 0.07
g/day for 5 days after hatching.

### 2.2. Survival to Pupation

One hundred and fifty *Ae. aegypti* eggs were hatched for each treatment and control. Each tray was inspected visually each day to count live pupae before eclosion, until no living larvae or pupae remained in the trays. This experiment was replicated six times for the 35 mg L^−1^ and four times for the 100 mg L^−1^ concentrations. Each replicate was coupled with a control. Differences in survival to pupation (SP) between control and treatments (henceforth expressed as “hazard ratios of pupation”) were analyzed through a Cox proportional hazards (CPH) model, including a cluster variable to control for the effect of individual replicates.

### 2.3. Wing Length

One hundred and fifty *Ae. aegypti* eggs were hatched for each treatment and control. Pupae were collected during the first three days after observation of the first pupa and transferred into a 200 mL cup placed in rearing cages. After 3 days, 10 males and females were sampled, cold killed and their right wing length (WL) measured under a dissecting microscope. Wing length was used as a proxy for the fitness of adult mosquitoes as it is closely associated with body size and nutritional reserves [24,25] and was measured as the distance from the alular notch to the apex of the wing, excluding the hairy fringe [26]. This experiment was replicated three times for the 35 mg L^−1^ and control concentrations. Two linear mixed models (LMM) were used to assess differences in WL between control and treatments for each sex. In each model, we used WL as the response variable, Rb or Cs concentration (i.e., Cs 35 mg L^−1^) as categorical covariates and replicate as a random factor. 95% CIs were estimated for all fixed model parameters and used to assess significance of differences in WL between control and treatments.

### 2.4. Adult Survival

One hundred and fifty *Ae. aegypti* eggs were hatched for each treatment and control. Pupae were collected during the first three days after observation of the first pupa and transferred into a 200 mL cup placed in rearing cages. To minimize the possibility of contamination of emerged adults through prolonged contact with Rb- or Cs-enriched larval water, the water cup was removed from the cage after 3 days. Adults were fed a 10% sucrose solution ad libitum using a 60 mL glass jar with a cotton wick inserted into the jar. The jar with cotton wick and sucrose solution was replaced every week until the end of the experiment. The number of dead adults was counted daily for 31 days from the day of first emergence. This experiment was replicated three times for the 35 mg L^−1^ and control concentrations. Differences in adult survival between control and treatments were analyzed through a Cox proportional hazards (CPH) model, including a categorical variable to control for the effect of replicates.

### 2.5. Detection of Rubidium and Caesium

One hundred and fifty *Ae. aegypti* eggs were hatched for 35 mg L^−1^ and control concentrations. Pupae collected during the first three days after observation of the first pupa were transferred into a 200 mL cup of DI water and placed in rearing cages. To minimize the possibility of contamination of emerged adults through prolonged contact with Rb- or Cs-enriched larval water, the water cup was removed from the cage after eclosion of the last pupa. Adults were fed a 10% sucrose solution *ad libitum* using a 60 mL glass jar with a cotton wick inserted into the jar. Adult females were offered defribinated sheep’s blood on two occasions, 6–8 and 12–14 days after eclosion. Females that were not engorged after the first blood meal were removed, thus the subsequent blood meal was provided only to adults that had completed a first gonotrophic cycle. One day after each blood meal, small plastic containers with DI water and seed germination paper were placed in each cage to supply the gravid adults with an oviposition substrate. The paper was removed and replaced with clean paper every day throughout the period of oviposition. This experiment was replicated three times for the 35 mg L^−1^ concentrations and control. A total of 20 adults 6–8 and 12–14 days old, as well as 20 eggs laid by adults 9–11 (first gonotrophic cycle) and 15–17 days old (second gonotrophic cycle) were taken from each treatment and stored in 2 mL vials for spectrometric analysis. Samples from the same treatment from different replicates were merged and stored at −20 °C.

Concentrations of Rb and Cs in both egg and adult samples were assessed through Inductively Coupled Plasma Mass Spectrometry (ICP-MS; 8900 ICP-QQQ, Agilent, Santa Clara, CA, USA) at the UC Davis Interdisciplinary Center for Inductively-Coupled Plasma Mass Spectrometry. ICP-MS allows the detection of metal and non-metal atoms in a sample with a resolution of parts per trillion ( 1 ng/kg). We introduced the samples in the ICP-MS using Laser Ablation (LA). LA is a technique for the semi-quantitative analysis of solids which makes use of UV lasers to ablate the sample, that is hence sent to the ICP as a gas. After the introduction of the gasified sample in the ICP-MS, the particles in the gas are progressively reduced to positively charged ions which are then analyzed in a quadrupole mass spectrometer. This analyzer separates ions based on their mass/charge ratio. Each ion that exits the quadrupole is then counted, producing a mass spectrum. The magnitude of each peak in the resulting mass spectrum is directly proportional to the concentration of an element in a sample.

Egg and adult mosquito samples were prepared in two different ways for ICP-MS analysis. Eggs were transferred initially from the germination paper on which they were laid to a 25×45
mm microscope slide using double-sided tape. The slide with the adherent eggs was placed on the translation stage of a New Wave Research UP-213 laser instrument. LA allows for highly targeted ablation and *Ae. aegypti* eggs, due to their elongated shape, were ablated along their longitudinal axis in an area 300 μm long and 80 μm wide (spot size aperture). Using this LA linear feature, each egg was uniformly sampled. Moreover, to ablate eggs without bursting, dislodging or reaching the tape below them, we set the laser power at 30% (~0.02 $J/cm^{2}$) with scan speed set to 5 μm/s. Before each scan, we allowed for 15 s laser warm-up time. As a result, the acquisition time (and length of each resulting mass spectrum) was 75 s. Between consecutive ablations, a wash-out period of 45 s was set in order to avoid cross-contamination between samples.

Adult females were extracted from storage vials with forceps and placed on double-sided tape adhered to a microscope slide. We decided to sample the scutum of adult females with LA since it is a reliably preserved and well-defined structure in field-collected mosquitoes. For the adult samples, the laser power was increased to 50% (~0.40 $J/cm^{2}$) to ablate the highly sclerotized cuticle of the scutum. The concentration of Rb and Cs in the gases contained in the LA translation stage chamber, herein referred to as background concentration, was also assessed to account for any potential environmental contamination.

The concentration of Rb and Cs in each sample was assessed considering its mass spectrum comprised between 15 and 45 s, as this part of the spectrum showed the lowest variability. The ICP-MS element analyzer returned element counts every 30.8 ms, thus each egg or adult female sample resulted in a total of 97 observations on Rb or Cs. We expressed Rb and Cs counts as a ratio respect to carbon counts (pRb and pCs) in order to correct for potential minor differences in laser ablation intensity between samples. Differences among samples were assessed considering the statistical distribution of corrected element counts. A sample was considered as successfully marked if its lower 95% CI pRb or pCs bound was higher than the upper 95% CI pRb or pCs of the control distribution.

All statistical analysis were performed in R 3.6.0 [27], the lme4 [28], survival [29] and survminer [30] packages were used to fit LMMs, perform CPH models and build forest plots, respectively.

## 3. Results

### 3.1. Pupation and Wing Length

The average SP in the control groups was 79.6% (Table 1). The treatment at 35 mg L^−1^ concentration showed no difference in the hazard ratio of pupation compared to controls, both for Rb and Cs (Figure 1). On the contrary, higher concentrations of both Rb and Cs resulted in decreased pupal survival. Overall, results revealed that at equivalent concentrations, larvae treated with Cs were exposed to a higher hazard of death before pupation than Rb. For an increase of Cs concentration from 35 to 100 mg L^−1^, the average hazard ratio of death increased from 0% to 82% for Cs and from 1% to 61% for Rb. Pupation did not occur in the 750 and 1000 mg L^−1^ treatments for Cs or in the 1000 mg L^−1^ treatment for Rb. The day of first pupation was delayed by one day (from day 5 to day 6 following egg submersion) at Cs concentrations 100 mg L^−1^ and higher and at Rb concentrations 500 mg L^−1^ and higher.

Wing lengths followed a pattern similar to that observed for SP (Figure 2). Model parameters showed that mosquitoes from control groups had an estimated WL of 3.01 mm and 2.37 mm, for females and males respectively (Table 1). Samples from 35 mg L^−1^ did not show significant differences in WL for Rb. On the contrary Cs at 35 mg L^−1^ reduced male WL by 0.16 mm but did not have a significant effect on females. One or both of the sexes showed a consistent decrease in WL for Rb or Cs concentrations of 100 mg L^−1^ or higher. At a Cs concentration of 500 mg L^−1^, the only emergent adult female had wings that were very short (1.5 mm) and deformed.

### 3.2. Adult Survival

The average adult survival after 31 days from first pupal eclosion was 49.6%, 56.7% and 62.2% respectively for control, Rb and Cs-treated group. There was no significant difference in the hazard ratio of death for adults between control and Cs, whereas the Rb-treated group showed a lower hazard of death than the control (Figure 3).

### 3.3. Detection of Rubidium and Caesium

Forty females (hereafter A-Rb35 and A-Cs35) and 40 eggs (hereafter E-Rb35, E-Cs35) from control and 35 mg L^−1^ Rb and Cs treatments were tested by mass spectrometry for the detection of Rb and Cs. The upper bound of 95% CI pRb in control adult samples was 0.03 (Figure 4A; Table 2). A-Rb35 consistently had a higher 95% CI pRb than controls; the lowest lower 95% CI pRb for a sample was 0.62. 12–14 day-old A-Rb35 (q0.50 = 6.66) had a slightly lower but not significantly different enrichment than 6–8 day-old A-Rb35 (q0.50 = 9.52). The upper bound of 95% CI pRb in control egg samples was 0.01 (Figure 4C; Table 2). Rb-treated eggs (E-Rb35) had a 95% CI pRb consistently higher than controls; the lowest lower 95% CI pRb recorded for a Rb-marked sample was 0.16. E-Rb35 from the second gonotrophic cycle (q0.50 = 2.26) had a slightly lower but not signficantly different enrichment than E-Rb35 from the first gonotrophic cycle (q0.50 = 6.22; Table 2).

The upper bound of 95% CI pCs in control adult samples was 0.01 (Figure 4B; Table 2). Eight out of 40 A-Cs35 had a lower bound 95% CI pCs higher than controls. All 8 samples were 12–14 days old A-Cs35. Overall, 12–14 day-old A-Cs35 (q0.50 = 0.11) had a lower enrichment than 6–8 day-old A-Cs35 (q0.50 = 0.64). The upper bound of 95% CI pCs in control egg samples was 0.01 (Figure 4D; Table 2). 6 out of 40 Cs-treated eggs (E-Cs35), all from the second gonotrophic cycle, had a 95% CI pCs lower than controls. Overall, E-Cs35 from the second gonotrophic cycle (q0.50 CI = 0.03) had a lower enrichment than E-Cs35 from the first gonotrophic cycle (q0.50 CI = 0.55; Table 2).

## 4. Discussion

In this study, we assessed the potential application of Rb and Cs as larval markers for *Ae. aegypti* in MR studies. By exposing immature mosquitoes to these elements during development, the handling of the dispersive adult stage is avoided, thus creating the potential for more accurate information to be gained on dispersal of marked mosquitoes. The proposed method has the added advantage of allowing for vertical transfer of the marks from females to their eggs, which establishes the potential for characterizing oviposition patterns in MR studies.

Results showed that a Rb concentration of 100 mg L^−1^ or greater reduced body size (WL) of adult mosquitoes, and increased immature mortality. Overall, Cs was more toxic than Rb, having a negative effect on male WL at just 35 mg L^−1^, which is likely to be due to the stronger ionic charge of Cs. Once Cs atoms have entered cells, they may be more difficult to remove due to their high affinity for negative charges in the intracellular fluid. Cs may therefore accumulate at high intracellular concentration, disrupting the osmotic equilibrium and the resting membrane potential of cells [8,31].

We did not observe any negative effect on immature and adult stages when larvae were reared at 35 mg L^−1^ Rb concentration. Therefore, adults and eggs from Rb and Cs 35 mg L^−1^ treatments were analyzed through spectrometry (although Cs at this concentration was associated with reduced male WL).

All adults and eggs tested for Rb reported Rb concentrations significantly higher than the concentration of control samples, regardless of the age of the sampled adults and the gonotrophic cycle at oviposition. This pattern did not hold true for Cs-treated samples, as eight 12–14 day-old females and 6 eggs from the second gonotrophic cycle showed no credible difference in Cs accumulation compared to controls. Moreover, median Rb abundance in marked samples showed a greater difference from the control median compared to Cs. A possible explanation for this result is that, given a potentially higher toxicity of Cs, only adult females with lower accumulated Cs survived through the second gonotrophic cycle.

Considering results from our fitness and survival experiments, together with results from spectrometric analysis, we conclude that Cs is not an optimal marker for *Ae. aegypti* MR studies. Even at low concentrations (35 mg L^−1^), Cs had a negative effect on mosquitoes and was not always detectable in exposed live individuals.

However, Rb at low concentrations (35 mg L^−1^) performed well as a marker. This element did not cause detectable effects on *Ae. aegypti* fitness and survival, and always accumulated at higher than natural levels in 2 week old females as well as in eggs through the second gonotrophic cycle. To the best of our knowledge, Rb is the only marker for mosquito MR experiments that has been shown to reliably transfer from exposed larval stages transstadially to adult females and then vertically to eggs. This characteristic of Rb enables the acquisition of additional data during MR experiments, namely the spatial and temporal patterns of oviposition by dispersing adult females [32]. Furthermore, the acquisition of this information does not hinder the collection of data on the dispersal of adult mosquitoes of both sexes, since both adults and eggs laid by exposed females are marked as part of the same process.

The applicability of Rb as marker at the larval stage has been tested previously for *Anopheles* mosquitoes [19]. These authors found that a water concentration of 100 mg L^−1^ was effective to mark *Anopheles* adult females up to ten days old. Our results showed that, even at a lower concentration of Rb, 12–14 old *Ae. aegypti* females retained a reliably detectable concentration of Rb. We cannot rule out physiological differences in Rb uptake and retention between *Anopheles* and *Aedes* genera, but much of the difference in results is probably attributable to the difference in sensitivity between the portable metal analyzer used in the earlier study compared to the more sensitive LA coupled with ICP-MS used in this study. Moreover, a more persistent Rb accumulation in our samples has likely been due to the earlier submersion of eggs into Rb-enriched water, instead of exposing the mosquitoes initially as third-instar larvae, as in reference [19]. These considerations should be taken into account in future MR experiments that utilize Rb as a marker.

In recent years, δ^13^C and δ^15^N have dominated MR studies for container-breeding mosquitoes (e.g., Hamer et al. [6,33], Opiyo et al. [34], Medeiros et al. [35]). Despite their many positive characteristics, isotopes and isotopic analysis are costly, their effectiveness depends on the quantity of the most abundant isotope in the environment and there is a relatively high risk of contamination. Moreover, artificially altered isotopic signatures are easily lost and thus undetectable in the next generation (eggs) due to the high physiological turn-over of C and N [6].

We have shown that Rb may be a complementary and more functional mosquito marker compared to stable isotopes of C and N. Rb is affordable and can be purchased from chemistry suppliers as RbCl crystals at a cost of around $50 per 10 g and, to obtain a concentration in water of 35 mg L^−1^, which was found optimal in our study, 0.049 g/L RbCl are required. Considering these figures, 10 g of RbCl are sufficient to mark *Ae. aegypti* in more than 200 L of larval water. Facilities that perform highly sensitive ICP-MS analysis are widespread and the cost of analysis is not excessive. We performed our analysis at the UC Davis Interdisciplinary Center for ICP-MS which applies a rate of $55 per hour for Supervised LA ICP-MS. The preparation of samples, especially if eggs, can be tedious initially but with some practice it becomes quite feasible and contamination is a relatively low risk under normal laboratory or field conditions.

## 5. Conclusions

The importance of detailed information on the dispersal of mosquito vectors is increasing [36]. Data on gene flow, range expansion and mating of released mosquitoes with wild populations are important for defining effective control strategies [37]. To date, MR experiments for mosquitoes remain time-consuming, expensive, and complex to implement, and technological advancements to overcome these challenges have been slow to materialize (but see Schmidt et al. [38]). Maximizing the quality and quantity of data acquired during MR experiments is therefore critical.

We found that Rb at a concentration of 35 mg L^−1^ in water with larvae did not affect the fitness or survival of *Ae. aegypti*, yet it was suitable to mark two-week-old adult females and eggs from the second gonotrophic cycle. The low cost of both Rb and ICP-MS analysis together with Rb’s minimal environmental toxicity make it a valuable marker for MR experiments with released or naturally occurring mosquitoes. On the contrary, Cs, even at low concentration, negatively affected *Ae. aegypti* fitness and did not consistently mark exposed individuals.

In conclusion, we recommend the use of Rb, alone or in combination with other markers, for mosquito MR experiments. We suggest that MR experiments for released or naturally occurring mosquitoes performed with stable isotopes of C and N could be enhanced by adding Rb as additional marker. In this way, data on the oviposition patterns would also be acquired and the number of possible marker combinations would increase from 3 to 6 (i.e., Rb+C, Rb+N, N, C, N+C, Rb), enabling marking of individuals from a higher number of larval habitats in the same MR experiment.

As a final note, despite the fact that Rb is not regarded as particularly toxic, it could have negative effects on the environment due to bioaccumuation and biomagnification [39]. We strongly advise against its application in natural habitats, for example in open water habitats such as wetlands.

## Figures and Tables

**Figure 1 insects-10-00220-f001:**
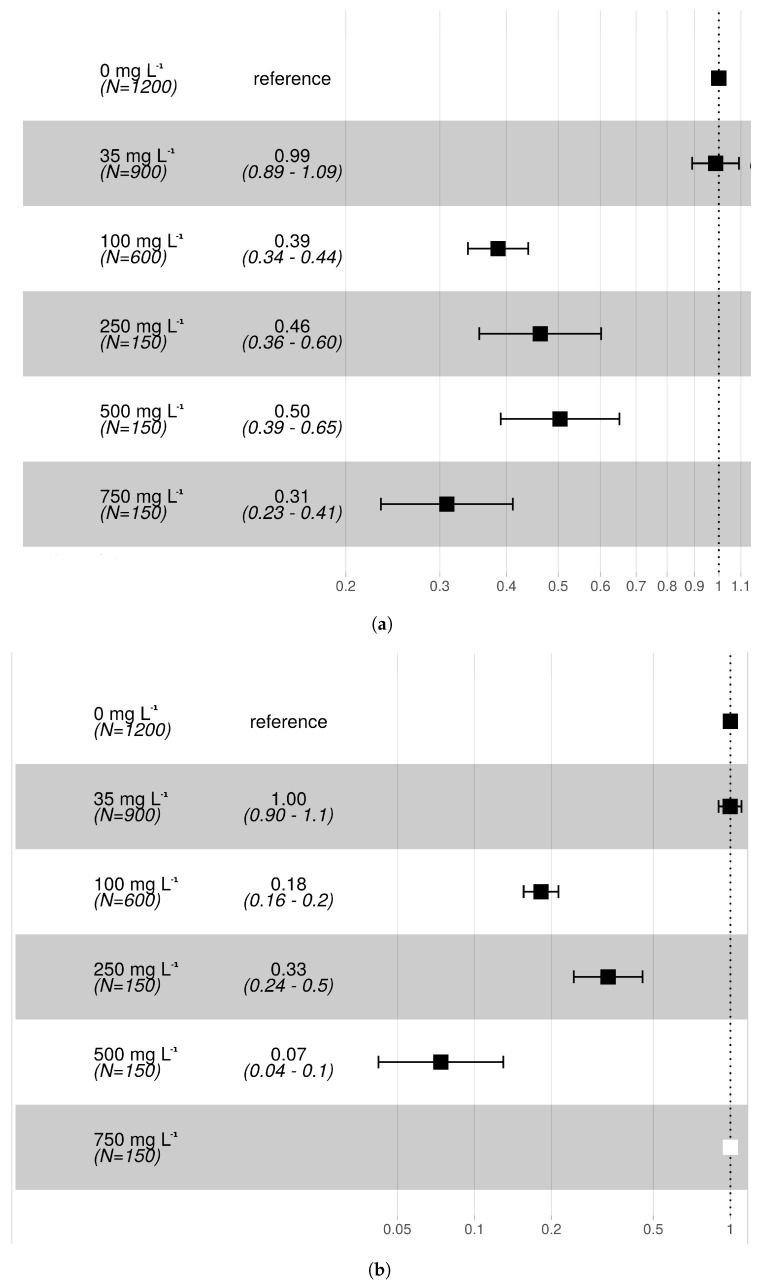
Forest plots reporting the hazard ratio of pupation for each (**a**) Rb and (**b**) Cs treatment with respect to the control (reference) group. The number of observations and average survival hazard ratios together with their 95% CI are shown as numbers in the second column or as squares and horizontal lines in the third column of each figure. The white square represents a treatment which did not show survival to pupa (i.e., Cs 750 mg/L).

**Figure 2 insects-10-00220-f002:**
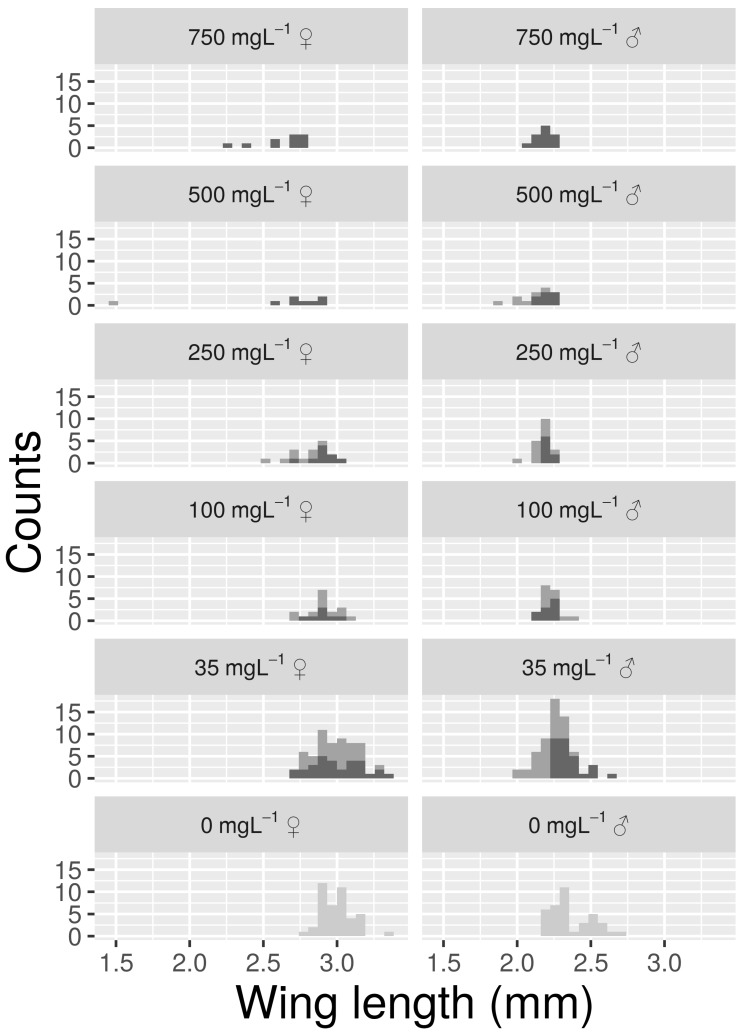
Histograms reporting the distribution of wing length (WL) values for the control (light grey) and for treatments (Rb = dark grey, Cs = grey) with at least one emerging adult.

**Figure 3 insects-10-00220-f003:**
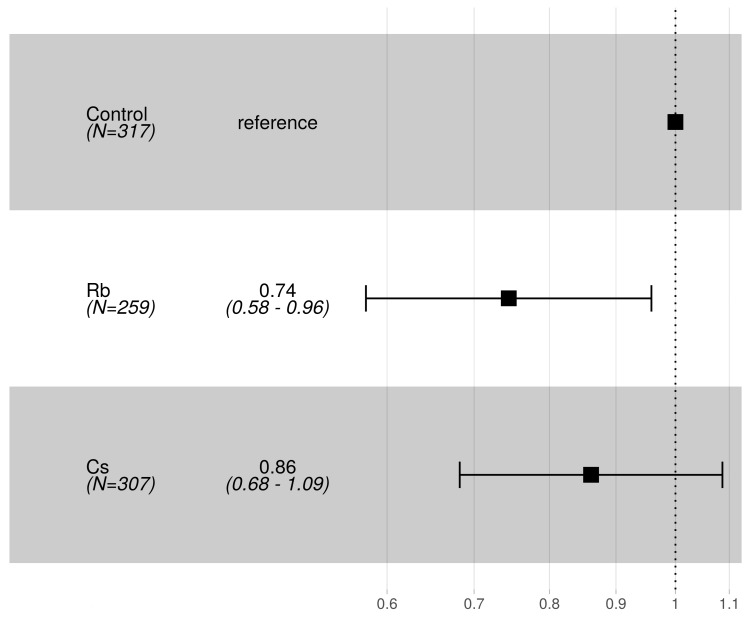
Forest plot reporting adult hazard ratios of death for Rb and Cs 35 mg L^−1^ treatments with respect to the control (reference) group. The number of observations and average hazard ratios of pupation together with their 95% CI are shown as numbers in the second column or as squares and horizontal lines in the third column of the figure.

**Figure 4 insects-10-00220-f004:**
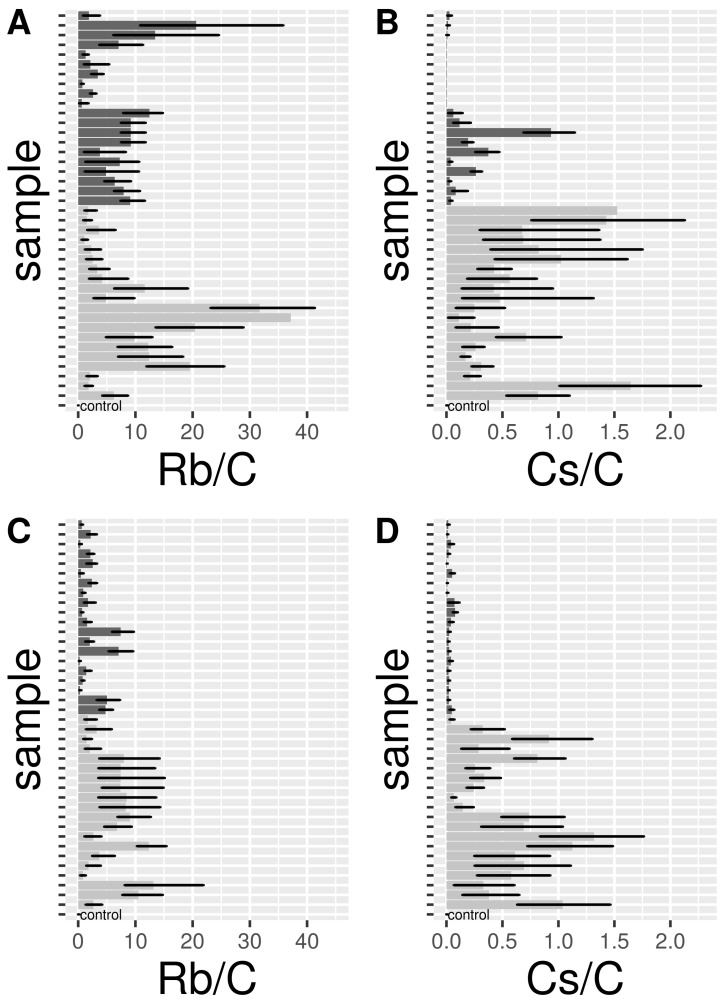
Barplots showing average and 95% CI bars for pRb (Rb/C, (**A**) = adults, (**C**) = eggs) and pCs (Cs/C, (**B**) = adults, (**D**) = eggs). Values for the controls are reported in the bottom bar of each figure. Light-grey bars show 6–8 days old adults (age-1) (**A, B**) or eggs (**C, D**) from the first gonotrophic cycle (gc-1), while dark-grey bars represent 12–14 days old adults (age-2) or eggs from the second gonotrophic cycle (gc-2).

**Table 1 insects-10-00220-t001:** Observed immature survival (%) and model parameter estimates for adult wing length (95% CI) for each treatment and the combined control group (0 mg L^−1^). For the two models, one for male adults and one for female adults, the standard deviation of the random effect “replicate” ($\sigma_{replicate}$) was 0.04.

Concentration (mg L^−1^)	Element	SP (%)	Model Estimated Wing-Length (mm)	
			Male	Female
0	-	79.6	2.37 (2.30, 2.43)	3.01 (2.94, 3.08)
35	Rb	82.4	−0.03 (−0.07, 0.02)	0.00 (−0.06, 0.06)
	Cs	78.9	−0.16 (−0.21, −0.12)	−0.02 (−0.08, 0.04)
100	Rb	71.2	−0.11 (−0.19, −0.04)	−0.06 (−0.17, 0.04)
	Cs	55.1	−0.08 (−0.16, −0.01)	−0.06 (−0.15, 0.02)
250	Rb	47.2	−0.11 (−0.20, −0.04)	−0.06 (−0.17, 0.03)
	Cs	33.7	−0.18 (−0.25, −0.11)	−0.21 (−0.31, −0.11)
500	Rb	51.0	−0.12 (−0.22, −0.06)	−0.18 (−0.29, −0.08)
	Cs	9.2	−0.29 (−0.38, −0.21)	−1.46 (−1.72, −1.21)
750	Rb	39.0	−0.15 (−0.22, −0.09)	−0.35 (−0.44, −0.26)
	Cs	0.0	-	-
1000	Rb	0.0	-	-
	Cs	0.0	-	-

**Table 2 insects-10-00220-t002:** Median (q0.50) and 95% CIs for pRb (Rb/C) and pCs (Cs/C), both for treatment and control groups. Age-1 = adults 6–8 days old, age-2 = adults 12–14 days old, gc-1 = eggs from the first gonotrophic cycle; gc-2 = eggs from the second gonotrophic cycle.

Element	Stage	Sample size	Group	q0.50	q0.25	q0.975
Rb	Adult	20	control	0.01	0.00	0.03
		20	age-1	9.52	1.00	38.02
		20	age-2	6.66	0.44	20.42
	Egg	20	control	0.00	0.00	0.01
		20	gc-1	6.22	0.82	15.03
		20	gc-2	2.26	0.23	7.99
Cs	Adult	20	control	0.00	0.00	0.01
		20	age-1	0.64	0.09	1.90
		20	age-2	0.11	0.00	0.96
	Egg	20	control	0.00	0.00	0.01
		20	gc-1	0.55	0.04	1.40
		20	gc-2	0.03	0.00	0.09

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
