# Peer review of "Revisiting Alkali Metals As a Tool to Characterize Patterns of Mosquito Dispersal and Oviposition"

_insects, 2019, doi:10.3390/insects10080220_

Round 1
Reviewer 1 Report
I have reviewed the manuscript titled "Revisting alkali metals as a tool for mark-release-recapture studies to characterise patterns of mosquito (Diptera: Culicidae) dispersal and oviposition" by Marcantonio et al. The manuscript is well written and provides useful insights to the appropriateness of these products for mosquito marking.
I do not have any major comments regarding the methodology, data collected, or analysis. However, there are just a couple of minor comments that the authors could consider including in a revised manuscript.
I would appreciate a brief justification from authors on why wing length measurement was used in this experiment (I could not find anything within the manuscript). Given it is typically a measure of larval nutrition or larval development times, what was the reasoning behind its inclusion here? A brief reasoning here would be useful within the context of this study but also for other researchers investigating the impact of alternative marking technologies in the future.
Does any research exist that points to any possible non-target impact of bioaccumulation of Rb (or Cs) in mosquito predators. While it could be argued that predation on Aedes aegypti may be minimal, many mark-release-recapture studies focus on much more abundant mosquitoes (e.g. wetlands associated Culex or Aedes spp) and any potential impact to birds/bats/predatory insects may need to be a consideration to other researchers. A brief comment, if possible, may be a useful addition.
Author Response
I have reviewed the manuscript titled "Revisting alkali metals as a tool for mark-release-recapture studies to characterise patterns of mosquito (Diptera: Culicidae) dispersal and oviposition" by Marcantonio et al. The manuscript is well written and provides useful insights to the appropriateness of these products for mosquito marking.
I do not have any major comments regarding the methodology, data collected, or analysis. However, there are just a couple of minor comments that the authors could consider including in a revised manuscript.
We thank the reviewer for their careful reading of the manuscript and their constructive remarks. We attached a new version of the manuscript which we modified following the reviewers' advice. We highlighted the changes in green, whereas strike-through text indicates parts to remove.
I would appreciate a brief justification from authors on why wing length measurement was used in this experiment (I could not find anything within the manuscript). Given it is typically a measure of larval nutrition or larval development times, what was the reasoning behind its inclusion here? A brief reasoning here would be useful within the context of this study but also for other researchers investigating the impact of alternative marking technologies in the future.
Wing length is a standard biometrics used as a proxy for body size in mosquito research. In this study, wing length was used to assess the fitness of adult mosquitoes reared in water medium with different concentration of Rb and Cs. Our working hypotheses was that smaller wing lengths were caused by a sub-optimal larval development due to toxic effect of Rb or Cs.
We added the following sentence in the text: “Wing length was used as a proxy for the fitness of adult mosquitoes as it is closely associated with body size and nutritional reserves (van Handel and Day 1988, van Handel and Day 1989), and was measured as the distance from the alular notch to the apex of the wing, excluding the hairy fringe (Packer and Corbet 1989)”
Does any research exist that points to any possible non-target impact of bioaccumulation of Rb (or Cs) in mosquito predators. While it could be argued that predation on Aedes aegypti may be minimal, many mark-release-recapture studies focus on much more abundant mosquitoes (e.g. wetlands associated Culex or Aedes spp) and any potential impact to birds/bats/predatory insects may need to be a consideration to other researchers. A brief comment, if possible, may be a useful addition.
Despite we did not find any research on the bioaccumulation of Rb in predators, we do not think that this is an issue for Rb-marked mosquitoes. Adults emerging from larvae reared at a Rb-concentration of 35 mg/L have a very small quantity of Rb in their tissue, possibly in the order of micrograms. It is therefore unlikely that the ingestion of rb-marked mosquitoes by predators will results in rb-bioaccumulation in their tissues.
However, we share the reviewer concern on the potential risk associated with bioaccumulation of rubidium in natural systems. We think that a greater risk of an impact to the environment may be posed by the utilization of Rb in natural habitats to study the dispersal of mosquitoes associated with open water habitats, such as salt-marsh or wetland mosquitoes. This type of marking would require large quantity of Rb in natural water habitats which may lead to biomagnification along the food chain with potential cascade effects on the ecosystem. To advise against the utilization of Rb in natural water habitats, we added the following sentence to the main text:
“As a final note, despite the fact that Rb is not regarded as particularly toxic, it could have negative effects on the environment due to bioaccumuation and biomagnification (see Campbell et al. 2005). We strongly advise against its application in natural habitats, for example in open water habitats such as wetlands or mashes.”
Reference list
Campbell, L. M., A. T. Fisk, X. Wang, G. Köck, and D. C. Muir. 2005. Evidence for biomagnification of rubidium in freshwater and marine food webs. Canadian Journal of Fisheries and Aquatic Sciences. 62: 1161–1167.
Handel, E. van, and F. Day Jonathan. 1988. Assay of lipids, glycogen and sugars in individual mosquitoes: correlations with wing length in field-collected Aedes vexans.
Handel, E. van, and J. F. Day. 1989. Correlation between wing length and protein content of mosquitoes. Journal of the American Mosquito Control Association. 5: 180–182.
Packer, M. J., and P. S. Corbet. 1989. Size variation and reproductive success of female Aedes punctor (Diptera: Culicidae). Ecological Entomology. 14: 297–309.

Reviewer 2 Report
The present manuscript revisits Cs- and Rb-enriched water as a method to mark Ae. aegypti mosquitoes and its progeny in order to apply the method for dispersal and oviposition studies. The manuscript is well-written and the findings are important and applicable. The methods are well-described as well. I consider this manuscript as " Accept after minor revision " after addressing a few questions and some modifications.
Below, the authors can find my considerations and suggestions:
1-The introduction section is too long and some paragraphs are redundant. For example, in line 83 ("the general objective...") and line 92 (The goal of....). These sentences would be condensed for better readability. "project"(line 92) could be replaced as well since it is not a project anymore.
2-Did the authors verify if the Rb could be sexually transferred to the mosquitoes? Would the authors be able to include this data in the manuscript? I would recommend this experiment. Or at least, the authors would comment on this specific point in their article.
3- In the case released RB-marked females bite humans, are there trace levels of Rb present in their saliva that would be inoculated in the host? Is it a potential risk to be addressed or it can be ignored? Could the authors include this experiment as well? I would recommend some commentary about this point in their discussion.
Author Response
The present manuscript revisits Cs- and Rb-enriched water as a method to mark Ae. aegypti mosquitoes and its progeny in order to apply the method for dispersal and oviposition studies. The manuscript is well-written and the findings are important and applicable. The methods are well-described as well. I consider this manuscript as " Accept after minor revision " after addressing a few questions and some modifications.
Below, the authors can find my considerations and suggestions:
We thank the reviewer for their careful reading of the manuscript and their constructive remarks. We attached a new version of the manuscript which we modified following the reviewers' advice. We highlighted the changes in green, whereas strike-through text indicates parts to remove.
1-The introduction section is too long and some paragraphs are redundant. For example, in line 83 ("the general objective...") and line 92 (The goal of....). These sentences would be condensed for better readability. "project"(line 92) could be replaced as well since it is not a project anymore.
We simplified the introduction making the last paragraph more concise and deleting repetitions.
2-Did the authors verify if the Rb could be sexually transferred to the mosquitoes? Would the authors be able to include this data in the manuscript? I would recommend this experiment. Or at least, the authors would comment on this specific point in their article.
In this study, we did not consider the possibility of sexual “transmission” of Rb as we focused on alkali metals as markers for mosquito MR studies. However, we demonstrated that Rb is transferred efficiently from marked females to their eggs and, likewise, we suppose that sperm are also Rb-enriched in male adults emerging from immatures exposed to Rb. It is improbable that the transfer of Rb-enriched sperm to not marked females leads to false positive detections of not marked females. Indeed sperm, after insemination, migrates in female’ spermathecae where it is maintained for a female's entire life. False positive detections of Rb-marked eggs due to enriched sperms are equally unlikely, due to the very limited numbers of sperms (1 to 10; Degner et al. 2016) entering mosquito eggs during egg fertilization.
While beyond the aim of this study, the point raised by the reviewer may be of interest for future projects researching mosquito sexual behaviors.
3- In the case released RB-marked females bite humans, are there trace levels of Rb present in their saliva that would be inoculated in the host? Is it a potential risk to be addressed or it can be ignored? Could the authors include this experiment as well? I would recommend some commentary about this point in their discussion.
Rb is regarded as a low toxic element for humans. This element is even used as an anti-depressant or “mineral supplement” and can be purchased in pharmacies or online shops without restrictions (see https://amzn.to/2KIKusi). The screening sub-chronic oral reference dose (RfD) of rubidium suggested by the USA Environmental Protection Agency (EPA) is 0.004 mg/kg of Rb per day, which equals to 0.28 mg of Rb/day for an average 70 kg adult person (EPA, 2016). Female mosquitoes emerging from immatures reared at a Rb-concentration of 35 mg/L will have absorbed a limited dose of Rb, possibly in the order of micrograms. The further presumable dilution of Rb in saliva, together with the very small quantity of saliva that a mosquito spits during an average blood meal (<1 nanoliter; Vogt et al. 2016), make human exposure to Rb due to bites by Rb-marked mosquitoes extremely unlikely. Therefore, we think that the possibility of health risks in humans due to Rb-marked mosquitoes should be considered very low.
Reference list
Degner, E. C., and L. C. Harrington. 2016. A mosquito sperm’s journey from male ejaculate to egg: Mechanisms, molecules, and methods for exploration. Mol Reprod Dev. 83: 897–911.
EPA. 2016. Provisional Peer-Reviewed Toxicity Values for Rubidium Compounds (CASRN 7440-17-7, Rubidium) (CASRN 7791-11-9, Rubidium Chloride) (CASRN 1310-82-3, Rubidium Hydroxide) (CASRN 7790-29-6, Rubidium Iodide).
Vogt, M. B., A. Lahon, R. P. Arya, A. R. Kneubehl, J. L. S. Clinton, S. Paust, and R. Rico-Hesse. 2018. Mosquito saliva alone has profound effects on the human immune system. PLOS Neglected Tropical Diseases. 12: e0006439.
